# Control of the temporal and polarization response of a multimode fiber

Mickael Mounaix[1]* & Joel Carpenter [1]

Control of the spatial and temporal properties of light propagating in disordered media have been demonstrated over the last decade using spatial light modulators. Most of the previous studies demonstrated spatial focusing to the speckle grain size, and manipulation of the temporal properties of the achieved focus. In this work, we demonstrate an approach to control the total temporal impulse response, not only at a single speckle grain but over all spatial degrees of freedom (spatial and polarization modes) at any arbitrary delay time through a multimode fiber. Global enhancement or suppression of the total light intensity exiting a multimode fibre is shown for arbitrary delays and polarization states. This work could benefit to applications that require pulse delivery in disordered media.

[1] School of Information Technology and Electrical Engineering, The University of Queensland, Brisbane, QLD 4072, Australia. *email: m.mounaix@uq.edu.au

Light propagation in disordered systems has been extensively studied over the past 40 years[1]. Transmitted light through such materials forms a complex pattern after propagation, known as speckle, that is the signature of the interference between a large number of modes within the sample. Although this speckle looks random, this mixing of light is nonetheless linear and thus deterministic. Wavefront shaping has revolutionized the spatial control of coherent light beams thanks to the use of spatial light modulators (SLM) in disordered systems[2,3], such as biological tissue, white paint or multimode fibers. Different approaches have been developed to control the propagation of coherent light that has experienced scattering, such as iterative optimization[4,5], optical phase conjugation[6], and the measurement of the optical transmission matrix[7]. These methods have been further extended to the control of polarization[8], spectral[9,10], and temporal properties of light[11], and widely used for imaging[12–15] and optical manipulation[16].

The spatial and temporal distortions of scattered light are coupled in disordered systems. Therefore, the temporal properties of a spatio-temporal speckle field can be adjusted with spatial-only control. In previous works, manipulating the temporal properties of scattered light was achieved with spatio-temporal focusing with a single SLM. The output pulse is focused on a single speckle grain, and the spectral properties of the focus are controlled to ensure a short duration of the output pulse. Spatio-temporal focusing of the output pulse can be reached for instance with optimization algorithms[17,18], digital phase conjugation[19], pulse shaping methods[20], or via the knowledge of the transmission matrix of the sample[11,21]. However, the output pulse has only a short duration at this specified focus position, while the background speckle remains temporally elongated[22]. The temporal control of the spatio-temporal field is then limited to a specific spatial mode (speckle position or a specific spatial pattern), at the expense of undefined temporal behavior for the other spatial modes. Equivalently, control of the impulse response of only a small portion of the total power means the remaining field is effectively a loss and potentially a source of noise. Most of the total input power does not arrive at a controlled delay, it is scattered into the temporal background. A control of the full speckle pattern over a certain spectral bandwidth has been proposed based on the transmission matrix, with the time delay operator[23–25]. Nonetheless, the limited controlled spectral bandwidth does not allow a full temporal control of the spatio-temporal speckle.

In this paper, we propose a novel method to adjust the temporal properties of the total output intensity at any arbitrary delay over all the spatial and polarization modes propagating through a multimode fiber. After experimentally measuring the Multi-Spectral Transmission Matrix (MSTM) of the fiber, we calculate the Time Resolved Transmission Matrix (TRTM) with a Fourier Transform. Exploiting the TRTM, we demonstrate for the first time both temporal enhancement and attenuation of the full beam at specific delay times at the expense of undefined behaviour in the spatial domain. In contrast with previous spatio-temporal focusing experiments where the temporal properties are adjusted only in the spatial focus position, all the output spatial modes are controlled at that specific delay. We also demonstrate control of the often neglected polarization degree of freedom, representing half of all the modes supported, which forms an unprecedented full control of the spatio-temporal output field over a large bandwidth.

## Results

**Experimental setup**. At a given wavelength $\lambda$, the transmission matrix $U$ of a disordered system, such as a multimode fiber, linearly links the input field (adjustable with a SLM) to the output field (detectable with a camera). $U$ is a $N_{\text{input}} \times N_{\text{output}}$ matrix, with $N_{\text{input}}$ and $N_{\text{output}}$ the number of spatial and polarization modes at the input and at the output of the system. Note that in a multimode fiber, all the $N_{\text{input}}$ modes (on the order of 100–1000 modes depending on its geometrical parameters) could be measured in contrast with highly disordered materials[26,27]. The transmission matrix has been widely used to manipulate light that has suffered from scattering or mode mixing, for focusing[7] or imaging purposes[28]. Recently, the transmission matrix has been extended to the spectral domain. The Multi Spectral Transmission Matrix[9,21], also known as the optical transfer function[29], is a stack of transmission matrices $U(\lambda)$ for a set of different input wavelength, that can be measured by sweeping the wavelength or using hyper-spectral imaging[30]. The MSTM enables a spectral control of the output field exploiting the spectral diversity of the medium, and could be used for focusing a pattern at a given wavelength[9,29], pulse shaping the output focus[21] and imaging through scattering samples[31].

We exploit the setup presented in Fig. 1 to measure the Multi Spectral Transmission Matrix of a multimode fiber (MMF). The light source is a tunable continuous wave (CW) laser (New focus TLM-8700) operating between 1510 nm and 1620 nm (bandwidth $\Delta\lambda = 110$ nm $\simeq 13.5$ THz). We use a phase-only spatial light modulator (Meadowlark P1920) to generate any spatial and polarization input state in amplitude and phase[32]. The SLM is located in the Fourier plane of a multimode fiber. The fiber is a 1-m length of a 50 $\mu$m core radius step-index multimode fiber (Thorlabs M42L01), which theoretically supports ~127 spatial modes per polarization state[33]. A step-index multimode fiber has

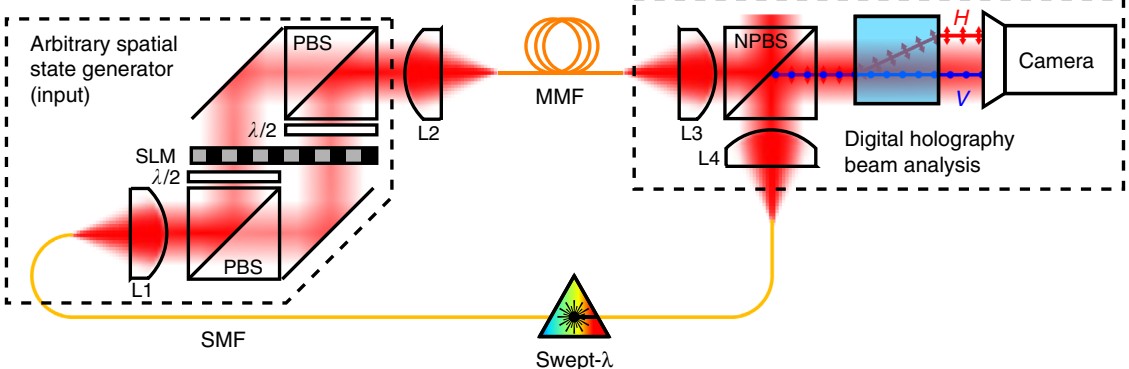

**Fig. 1** Apparatus of the experimental setup. SMF: single mode fiber; SLM: spatial light modulator; PBS: polarizing beam splitter; MMF: multimode fiber; NPBS: non-polarizing beam splitter; (L1, L2, L3, and L4): optical lenses. Focal length: L1: 25 mm; L2: 10 mm; L3: 4 mm; and L4: 60 mm

more mode coupling and a more complex and spread impulse response than a graded index fiber with similar dimensions. The multimode fiber we use in this experiment is not in the strong mode coupling regime[34] (see Supplementary Note 2 for more details on the mode coupling in the fiber). The output field is measured with digital off-axis holography, using an external path-length matched reference beam and an InGaAs camera (Xenics Xeva-1.7-320). The two polarization states are measured on the left and the right side of the camera using a polarizer beam displacer.

**Measurement of the Transmission Matrix of a multimode fiber.** We use a swept-wavelength interferometer (SWI) to measure the multi spectral transmission matrix of the MMF[29,32]. We measure the complete transmission matrix by generating all the spatial and polarization modes (a total of $N_{\text{input}} = 254$ modes) on the input facet of the MMF using the SLM. We launch the modes of the step-index fiber in a linearly polarized basis with helical wavefronts. For each individual input mode, we sweep the laser and record the output field for all the wavelengths with the digital holography beam analysis. Through the sweep, the camera is triggered at 10.6 GHz through the sweep with an external Mach-Zehnder interferometer that acts as a k-clock. This triggering method enables the measurement of $N_{\lambda} = 1273$ monochromatic transmission matrices per sweep. Each output field is then digitally decomposed on the fly in the Laguerre-Gaussian basis ($N_{\text{output}} = 650$ modes). For a specific input mode, the corresponding 2D slice of dimension $N_{\text{output}} \times N_{\lambda}$ of the MSTM is measured in ~16 seconds: the full MSTM is then measured in a bit more than an hour. Additional details on the experimental setup are provided in Supplementary Note 1.

**Enhancing the intensity at an arbitrary delay time.** With a sufficient spectral sampling, the MSTM can be Fourier transformed to access the transmission matrices as a function of delay $U(\tau)$. The obtained Time Resolved Transmission matrix (TRTM) contains the full relationship between the spatiotemporal input field to the spatiotemporal output field. The TRTM has been exploited so far to focus light in space and time[11] and to study spatio-temporal correlations[35]. We show in Supplementary Fig. 1 a set of transmission matrices at different delay times of the multimode fiber, that we calculated from our measured MSTM.

We demonstrate here how to exploit the TRTM to control the light intensity integrated over all spatial and polarization modes at an arbitrary delay $\tau_s$. The transmission matrix of the MMF at $\tau_s$, that we write $U_{\tau_s}$, is extracted from the TRTM. Initially developed in acoustics[36,37], the Time Reversal Operator $U^{\dagger}U$, where the $\dagger$ operator stands for the conjugate transpose, has been extensively used in the monochromatic regime for selective focusing[38], and the study of open channels[39–42]. Indeed, its eigenvalues are directly related to the total transmitted energy at the output of the disordered system[26,43]. In the time domain, the eigenvectors of $U_{\tau_s}^{\dagger}U_{\tau_s}$ have been used to study optimal transmission through scattering systems[44]. For instance, in order to maximize the transmitted energy at the delay time $\tau_s$, the input state $E_{\text{input}}^{\max}$ is the eigenvector of $U_{\tau_s}^{\dagger}U_{\tau_s}$ associated to the highest eigenvalue[35]. Our experimental setup enables us to measure all spatial modes in both polarizations at both ends of the fiber within $U(\lambda)$, and thus within $U_{\tau_s}$. Therefore launching $E_{\text{input}}^{\max}$ is thus enhancing both the horizontal and the vertical polarization states at the output of the MMF at the delay time $\tau_s$. In contrast with digital phase conjugation of $U_{\tau_s}$[11], this eigenstate is not focusing light in a certain spatial mode or in a spatial position: it

is enhancing the light intensity at $\tau_s$ for all the spatial and polarization modes.

We experimentally demonstrate this enhancement of light intensity at an arbitrary delay time in Fig. 2. The delay time $\tau = 0$ ps corresponds to the arrival time of the first principal mode of the MMF[23]. As an illustrative example, we maximize the transmitted intensity at $\tau_{\max} = 8.5$ ps in Fig. 2a. Once we display the eigenvector associated to the maximum eigenvalue of $U_{\tau_{\max}}^{\dagger}U_{\tau_{\max}}$ onto the SLM, we measure the impulse response $I(\tau)$ of the output state. For this purpose, we measure the spectrally resolved output field $E_{\text{output}}(\lambda)$ with the SWI, while the fixed input state is being displayed on the SLM. The impulse response is calculated from the superposition over all the $N_{\text{output}}$ output modes:

$$I(\tau) = \sum_{j=1}^{N_{\text{output}}} |\hat{E}_{\text{output}}(\tau, j)|^2 \quad (1)$$

with $\hat{E}_{\text{output}}(\tau, j)$ the Fourier transform of $E_{\text{output}}(\lambda)$ decomposed on the $j$-th output mode. The impulse response is then normalized. The calculated impulse response with Eq. 1 is then equivalent to what would be measured by a large photodetector at the end of the fiber.

In Fig. 2a, we plot the experimentally measured time of flight distribution of the multimode fiber in blue color as a reference impulse response. It corresponds to the average impulse response $I_{\text{rand}}$ over a set of $N_{\text{rand}} = 20$ random combinations over all spatial and polarization input states. The experimentally measured impulse response $I_{\tau_{\max}}$ of the maximum eigenstate at $\tau_{\max}$ is plotted in red in Fig. 2a, which clearly shows a sharp peak at $\tau_{\max}$. This impulse response could be numerically predicted with the MSTM. For every wavelength $\lambda$, the expected output field $E_{\text{output}}^{\text{numerical}}(\lambda)$ reads $E_{\text{output}}^{\text{numerical}}(\lambda) = U(\lambda) E_{\text{input}}^{\max}$. From the set of $E_{\text{output}}^{\text{numerical}}(\lambda)$, we can use Eq. 1 to calculate the expected impulse response $I_{\tau_{\max}}^{\text{numerical}}$. We plot $I_{\tau_{\max}}^{\text{numerical}}$ in green in Fig. 2a, which is, as expected, very similar to the experimentally measured impulse response.

From the spectrally resolved field measurements, we can reconstruct the output intensity at the delay time $\tau_{\max}$ with a Fourier transform. In the inset of Fig. 2a, we show the reconstructed intensity at $\tau_{\max}$ for both polarization states for the enhancing eigenstate (top row). The output intensity is here spread over a combination of spatial modes for both the horizontal and the vertical polarization. As a contrast with digital optical phase conjugation of the TRTM[11] and previous spatio-temporal control through scattering systems in the literature, the output intensity is not spatially focused to a mode or to a speckle grain, as all the output spatial modes are enhanced at that specific delay. As a comparison, we also show the reconstructed intensity of a random combination of input modes of both polarization at $\tau_{\max}$, with the same color-bar (bottom row).

Similarly to spatial focusing experiments[4], we define an enhancement ratio $\eta(\tau_{\max})$ when enhancing the output intensity at delay time $\tau_{\max}$. Based on the impulse responses plotted in Fig. 2a., $\eta(\tau_{\max})$ is defined as follows:

$$\eta(\tau_{\max}) = \frac{I_{\tau_{\max}}(\tau_{\max})}{I_{\text{rand}}(\tau_{\max})} \quad (2)$$

with $I_{\tau_{\max}}(\tau_{\max})$ and $I_{\text{rand}}(\tau_{\max})$ the intensity of the impulse response of the enhancing eigenstate and the time of flight at the delay time $\tau_{\max}$. In Fig. 2b, we compare the enhancement ratio $\eta(\tau)$ calculated with the experimentally measured impulse

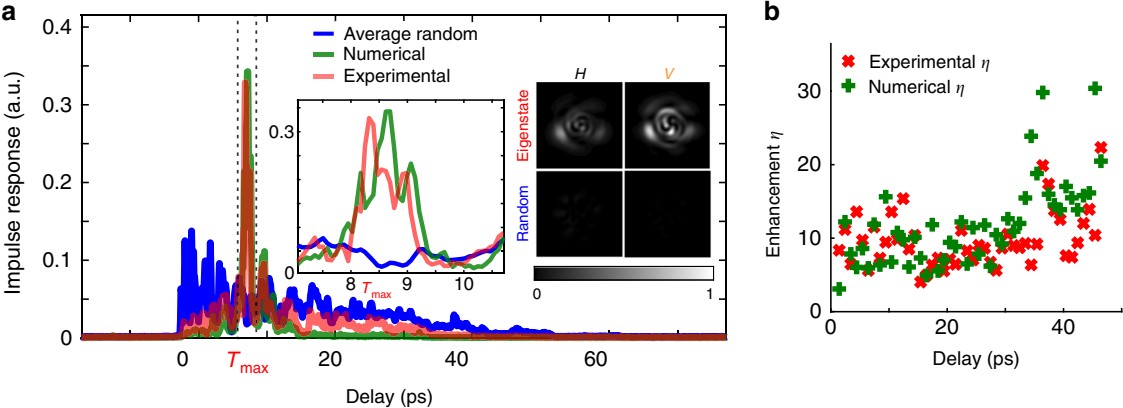

**Fig. 2** Enhancing light intensity at a chosen delay time over all the spatial and polarization modes of a multimode fiber. **a** Enhancing light intensity at $\tau_{max} = 8.7$ ps. Blue: time of flight distribution of the multimode fiber. Red: experimental impulse response of the eigenstate maximizing the light intensity at $\tau_{max}$. Green: numerical propagation of this eigenstate with the experimentally measured MSTM. Inset: zoom around $\tau_{max}$; Reconstructed output intensity at $\tau_{max}$ for the two polarization states for the eigenstate (top row) and a random combination of modes (bottom row). **b** Enhancement ratio $\eta$ for different delay times. Red: enhancement ratios calculated with the experimentally measured impulse responses. Green: enhancement ratios calculated with the numerically propagated impulse responses

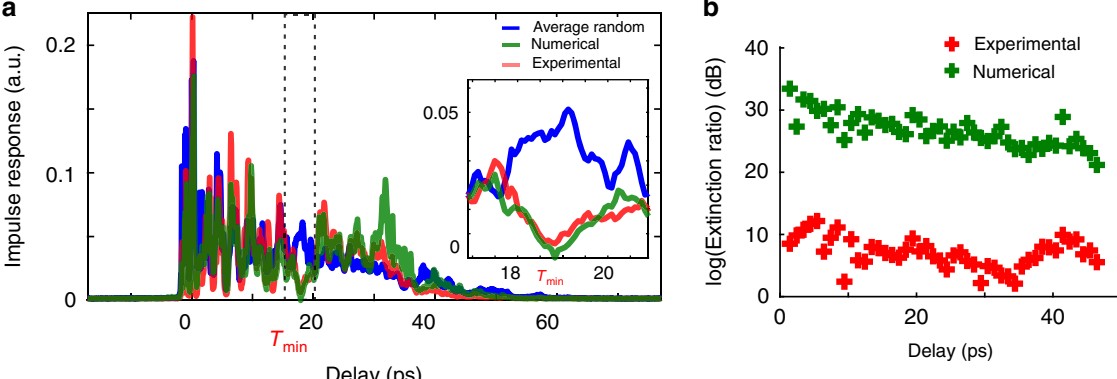

**Fig. 3** Attenuating light intensity at a chosen delay time. **a** Attenuating light intensity at $\tau_{min} = 19$ ps. Blue: time of flight distribution of the multimode fiber. Red: experimental impulse response of the eigenstate minimizing the light intensity at $\tau_{min}$. Green: numerical propagation of this eigenstate with the experimentally measured MSTM. Inset: zoom around $\tau_{min}$ (**b**) Attenuation ratio $\rho$ for different delay times in logarithm scale. Red: attenuation ratios calculated with the experimentally measured impulse responses. Green: attenuation ratios calculated with the numerically propagated impulse responses

responses (red markers) with the expected enhancement $\eta^{numerical}(\tau)$ (green markers) for a set of different arrival times spread between $\tau = 0$ ps and $\tau = 45$ ps. The expected enhancement can be predicted with the TRTM. However, we cannot use the random matrix theory to predict the eigenvalue distribution as the MMF is not in the strong mode coupling regime[35]. The expected enhancement $\eta^{numerical}(\tau)$ is calculated with Eq. 2, where $I_{\tau_{max}}^{numerical}$ replaces $I_{\tau_{max}}$. As the TRTM contains the full spatiotemporal relationship between the input and the output fields, $\eta(\tau)$ follows, as expected, a similar trend as $\eta^{numerical}(\tau)$. Modes arriving at very long delays are higher-order modes, that are near cut-off. These higher-order modes experience less mode coupling and hence sharper temporal features[45], which tends to increase the enhancement ratio for the eigenmodes at long delays. Additional data processing, such all the individual impulse responses used to calculate $\eta$ and the spatial reconstruction of the beam at the different delays, are presented in the Supplementary Note 4. Finally, in our study we do not launch an input pulse as we measure the data with swept-wavelength digital holography. A simulation of the propagation of an input pulse, and its effect on the impulse response and the enhancement is presented in the Supplementary Note 4.

**Attenuating the intensity at an arbitrary delay time.** The time reversal operator $U_{\tau_s}^{\dagger} U_{\tau_s}$ enables to adjust the transmitted intensity over all spatial and polarization modes at the delay $\tau_s$. Exploiting the eigenvector associated with its maximum eigenvalue enables enhancement of intensity at $\tau_s$. Instead of enhancing the light delivery, we can also attenuate the impulse response at the delay time $\tau_s$ with the eigenvector $E_{input}^{min}$ associated with the minimum eigenvalue of $U_{\tau_s}^{\dagger} U_{\tau_s}$. We illustrate in Fig. 3a the experimental impulse response associated with the minimum eigenstate at a chosen delay time $\tau_{min} = 19$ ps. The expected impulse response shows a clear minimum intensity at $\tau_{min}$. Experimentally, the impulse response reveals as well a sharp minimum at $\tau_{min}$, whose intensity is much smaller than the time of flight distribution at the same arrival time.

Similarly to the enhancement ratio defined in Eq. 2, we also define an attenuation ratio $\rho(\tau_{min})$ for such minimum eigenstates:

$$\rho(\tau_{min}) = \frac{I_{rand}(\tau_{min})}{I_{\tau_{max}}(\tau_{min})} \quad (3)$$

We plot the attenuation ratio in logarithm scale for a set of different delay time between 0 ps and 45 ps in Fig. 3b. The

numerical extinction ratio <− 20 dB implies we should be able to completely attenuate the beam at every delay time, as the inset of Fig. 3a illustrates for instance at specific delay $\tau_{\min}$. However, the experimental conditions are not allowing a pure zero-intensity in the experimentally measured impulse response. Indeed, the MSTM of the MMF has slightly decorrelated between its measurement and when the eigenstates are being measured. Besides, the superposition of the $N_{\text{input}} = 254$ input modes on the SLM might not be as accurate as expected in the experiment. Finally, the experimental stability level, whether on the detection as well as on the illumination, add some noise which prevents the observation of a pure zero-intensity in the impulse response. Ultimately, this non-zero experimental intensity tends to decrease the extinction ratio at larger delays where the time of flight gets a smaller intensity.

**Polarization control of the generated output state.** The propagating modes in the multimode fiber are polarization-resolved. In order to address all the modes, it is essential to control both the spatial and the polarization degrees of freedom. Wavefront shaping experiments in disordered materials, including multimode fiber, are often sensing only a quarter of the transmission matrix, by neglecting a polarization state both at the input and at the output[7,24]. Neglecting the polarization is equivalent to ignoring half of the propagating modes in the fiber. Full control of all the modes requires then the measurement of the polarization-resolved transmission matrix of the fiber. The experimental setup presented in Fig. 1 enables the measurement of the polarization-resolved MSTM of the MMF. Indeed, we can generate and detect independently the two polarization states for all the spectral components. Therefore the MSTM $U(\lambda)$ could be written as follows:

$$U(\lambda) = \begin{bmatrix} U_{HH}(\lambda) & U_{HV}(\lambda) \\ U_{VH}(\lambda) & U_{VV}(\lambda) \end{bmatrix} \quad (4)$$

with $U_{ij}(\lambda)$ the transmission matrix relating the input spatio-temporal field on the $i-th$ polarization state to the output spatio-temporal field on the $j-th$ polarization state. The dimension of each sub-matrix is $N_{\text{input}}/2 \times N_{\text{output}}/2$. Let's assume we want to generate an arbitrary state in a chosen polarization state $j$ at the output of the fiber at a wavelength $\lambda$, whether $j$ is H or V. For this purpose we thus use $U_j(\lambda) = [U_{Hj}(\lambda) \, U_{Vj}(\lambda)]$[29]. In the delay domain, the calculated TRTM from the experimentally measured MSTM is also polarization-resolved: $U_{j,\tau_s}$ is the TRTM at the delay time $\tau_s$ for the output polarization state $j$. The time reversal operator $U_{j,\tau_s}^{\dagger} U_{j,\tau_s}$ enables then a temporal control of the overall combination of spatial modes for a specific arrival time $\tau_s$ and a specific polarization state at the output of the MMF.

Similarly to Fig. 2, the output intensity can be enhanced over all the spatial modes for only the $j$-polarization state at the output. The input field $E_{\text{input}}^{\max,j}$ is the eigenvector associated to the maximum eigenvalue of $U_{j,\tau_s}^{\dagger} U_{j,\tau_s}$. In Fig. 4a, b, we plot the polarization-resolved impulse responses associated to such eigenstates with $\tau_s = 8.5$ ps. The polarization-resolved impulse responses are calculated with Eq. 1, using only the output modes on the chosen polarization state. Figure 4a, b clearly evidence an enhancement of the intensity overall spatial modes at $\tau_s$ for either only H or only V. The reconstructed intensity at the output for H and V at the delay time $\tau_s$ confirms that the transmitted light at $\tau_s$ is polarized in the chosen state. An incomplete control of the input polarization state would lead to similar results, but with a lower energy delivery at $\tau_s$ as half of the input power would not

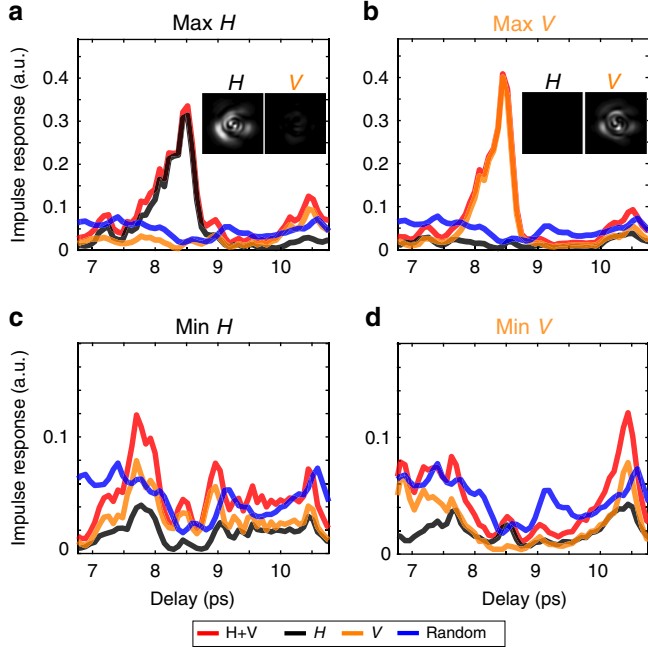

**Fig. 4** Polarization control of the engineered state at an arbitrary delay time $\tau_s$. We choose here $\tau_s = 8.5$ ps. **a** Enhancing only the H component of the output field. **b** Enhancing only the V component of the output field. **c** Attenuating only the H component of the output field. **d** Attenuating only the V component of the output field. Inset: reconstructed intensity at the output at delay time $\tau_s$ for both H and V. Color code: (red) total impulse response of H and V; (black) impulse response of the H component; (orange) impulse response of the V component; (blue) time of flight distribution of H and V

be controlled. Such an incomplete polarization control is shown in Supplementary Fig. 8. The MMF in conjunction with the SLM can then be used as an accurate deterministic polarization control device in the time domain. This experiment could also be used to study polarization recovery in scattering media[46].

We also demonstrate an attenuation of a chosen polarization state at the output of the fiber at the delay time $\tau_s$. Instead of launching $E_{\text{input}}^{\max,j}$, we calculate $E_{\text{input}}^{\min,j}$ which is the input eigenvector associated to the minimum eigenvalue of $U_{j,\tau_s}^{\dagger} U_{j,\tau_s}$. Figure 4c, d present the experimental polarization-resolved impulse responses associated to such eigenstates for attenuating either only H or only V. Figure 4c, d unambiguously reveal a strong attenuation at $\tau_s$ for the controlled output polarization state. In principle, any arbitrary polarization state can be achieved at a given delay, similarly to previous work in the monochromatic regime[47], based on the superposition of the four sub time gated transmission matrices of Eq. 4. Some examples are shown in Supplementary Fig. 9.

**Multiple delay control.** In Fig. 2, we have demonstrated the enhancement of light intensity at a chosen arbitrary time $\tau_s$ over all the spatial and polarization modes. The TRTM could also be used to shape more complex impulse responses. Spatio-temporal focusing over multiple space-time positions have been demonstrated with digital optical phase conjugation of the TRTM[11]. Here we demonstrate an enhancement of the transmitted light intensity at multiple delay times with the experimental setup of Fig. 1.

The time reversal operator $U_{\tau_s}^{\dagger} U_{\tau_s}$ enables to adjust the transmitted energy at the delay time $\tau_s$. Figs. 2 and 3 have shown this control for either enhancing or attenuating light intensity at a

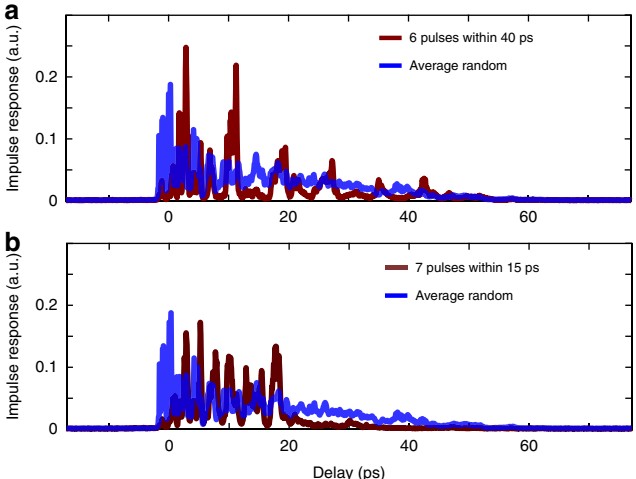

**Fig. 5** Enhancement of transmitted light intensity at multiple delay times. **a** Enhancing the light intensity at 6 different delays $\tau_i$ evenly spread from 4 ps to 44 ps. **b** Enhancing the light intensity at 7 different delays $\tau_i$ evenly spread from 4 ps to 19 ps

chosen delay time $\tau_s$. The spectral diversity of the multimode fiber ensures that this control is only affecting delays around $\tau_s$, as the transmission matrices as a function of wavelength/delay are uncorrelated over a sufficient bandwidth/time interval[21,23,24].

In order to enhance light intensity at two different delays $\tau_1$ and $\tau_2$, we use the time reversal operators $U_{\tau_1}^\dagger U_{\tau_1}$ and $U_{\tau_2}^\dagger U_{\tau_2}$. We calculate the input eigenvectors $E_{input}^{max,\tau_1}$ and $E_{input}^{max,\tau_2}$ associated to the highest eigenvalue of the two single operators. The linear superposition $E_{input}^{max,\tau_1,\tau_2} = E_{input}^{max,\tau_1} + E_{input}^{max,\tau_2}$ enables a control of the impulse response at the two delay time $\tau_1$ and $\tau_2$. The phase of the solution would then be displayed on the SLM, similarly to spatio-temporal focusing at multiple delays[11,21]. This control can be extended to a superposition of $N_{superpos.}$ delay times. The input field would then read:

$$E_{input}^{max,N_{superpos.}} = \sum_{i=1}^{N_{superpos.}} E_{input}^{max,\tau_i} \qquad (5)$$

with $E_{input}^{max,\tau_i}$ the eigenvector associated with the highest eigenvalue of $U_{\tau_i}^\dagger U_{\tau_i}$. In practice, $N_{superpos.}$ would be limited by the temporal width of the impulse response of each single mode. The impulse responses may interfere for different times, especially if the $\tau_i$ are close in the time domain. We plot in Supplementary Fig. 3 the temporal correlation between the different Time Resolved Transmission Matrices. The quality of the superposition of the modes on the SLM would also limit the efficiency of a high $N_{superpos.}$.

In Fig. 5 we demonstrate multiple delay control of the transmitted intensity with the experimental setup of Fig. 1. In Fig. 5a, $N_{superpos.} = 6$ delay times are controlled, evenly spread between 4 ps and 44 ps. Figure 5b illustrates a similar control over a narrower time interval. The intensity of $N_{superpos.} = 7$ delay times is enhanced, evenly spread between 4 ps and 19 ps. We clearly identify $N_{superpos.}$ peaks in Fig. 5a and Fig. 5b. The enhancement ratios at the $N_{superpos.}$ delays are much lower than the individual delay enhancement ratios presented in Fig. 2b. However the total transmitted power remains similar for either a single delay enhancement or a multiple delay control. The amplitude ratio between these peaks could also be adjusted using an amplitude coefficient in the superposition in Eq. 5. A control

of different $N_{superpos.}$ within the same time interval is presented in the Supplementary Figs. 10 and 11. This multiple delay control could also be exploited in conjunction with Fig. 4, where the two polarization states could be manipulated at different arrival times.

The ability to manipulate multiple delay times within long delay and dispersion provided by a very compact optical system such as a multimode fiber opens interesting perspectives for compact pulse shaping experiments. We believe this multiple delay control might also be useful for coherent control and pump probe experiments for imaging purposes or for complex control of light-matter interactions.

## Discussion
Over the last decade, wavefront shaping with spatial light modulators have revolutionized the study of light propagation in disordered systems, where scattering was considered as an insurmountable obstacle. Most of the previous works studied spatial control of the output field at the expense of the temporal behavior. We have demonstrated here the temporal control of light intensity averaged over all the spatial and polarization modes after propagation through a multimode fiber, at the expense of the spatial pattern. More precisely, we have shown how to enhance or attenuate the transmitted intensity at any arbitrary delay times, arbitrary polarization state, and arbitrary combinations of these. This work opens interesting perspectives in applications that require the measurement or control of spatio-temporal properties of light in scattering media. A pulse with a large number of transverse modes could be useful for the following non exhaustive list of examples: the study of correlations in disordered systems[35,48–50] telecommunications[51], non-linear imaging[31,52] and endoscopy[53–55] through complex systems, time-gating imaging[56,57], temporal focusing microscopy[58–60] and high power fiber lasers[61]. Our polarization control of the beam in the temporal domain could find applications in polarization sensitive optical coherence tomography[62–64]

## Acknowledgements
We thank Martin Plöschner for careful reading of our manuscript. This work is supported by the Australian Research Council (ARC) projects DE180100009 and DP170101400.

## Data availability
The data that support the findings of this study are available from the corresponding author upon reasonable request.

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

## Author contributions

M.M. initiated the study and performed the experiments. M.M. and J.C. led the study, contributed to data analysis and writing the manuscript.

## Competing interests

The authors declare no competing interests.
