## [Peer Review File · Nature Communications]

Reviewers' Comments:

Reviewer #1:

Remarks to the Author:

In this manuscript the authors demonstrated a full control of transmitted light intensities in all spatial and polarization modes of a multimode fiber. The multi-spectral transmission matrix was measured and a Fourier transform gave the time-resolved transmission matrix. The eigenstates of the time-resolved transmission matrix with the maximum/minimum eigenvalues were computed and launched into the fiber for temporal enhancement or attenuation of transmitted light at a specific delay time.

This method was already used in Ref. [19] for global focusing of light through a multimode fiber. The main improvement in this work is that both polarizations are measured at input and output of the fiber. Similar to Ref. [19], the temporal enhancement was shown indirectly by Fourier transform of the output field measured at each frequency. It would be more impressive if the authors could directly measure the temporal pulse shape and show the temporal enhancement/attenuation, similar to what one of the authors had done for spatio-temporal focusing through disordered systems in Ref. [21].

Below are my questions.

1. How strong were the mode coupling and the polarization mixing in the step-index fiber used in the experiment?
2. What determined the enhancement factor in Fig. 2b? Why it increased with the delay time? How did the enhancement factor depend on the fiber parameters and the input pulse width?
3. In Fig. 3b, why the extinction ratio decreased with increasing delay time, opposite to the enhancement factor in Fig. 2b?
4. Figure 4 showed either horizontally or vertically polarized light was enhanced or suppressed at a specific delay time. Can the authors demonstrate arbitrarily polarization state, e.g., circular polarization, at a specific delay time, similar to what was done with continuous wave by Xiong et al. in *Light: Science & Applications* 7, 54 (2018)?
5. When the transmitted light intensity is enhanced at multiple delay times in Fig. 5, how much was the total enhancement factor? Was it lower than the enhancement factor at a single delay time?

In summary, the experimental studies presented in this paper were solid, but physical understanding of the experimental results was missing.

Reviewer #2:

Remarks to the Author:

Control of the temporal and polarization response of a multimode fiber

This article describes the control of the temporal shape and polarization of the total transmission through a multimode fiber. This is in contrast to typical pulsed experiments in which the pulsed intensity is controlled in a single speckle spot in transmission through a scattering medium. In that case, the total transmitted power is dominated by the background, which is enhanced to a lesser degree than the intensity in the focal spot. Here the total transmission over the full core of the fiber is shaped. In steady state as well as in pulsed experiments, the ratio between transmission in the highest and lowest transmission eigenvalues is determined by the completeness of the

transmission matrix [S M Popoff, PRL, 112, 133903,2014; A Goetschy PRL 111, 063901 2013]. Apparently, the matrix is quite complete here giving a large ratio of maximum to minimum intensity at a specified time. To accomplish this it is important to have full control over the polarization. A detailed discussion of the extent and sources of incompleteness of the transmission matrix in this case would be of interest. Unlike the transmission through a scattering medium, it appears that all channels are completely transmitted and so the number of channels is just the number supported by the fiber. By dividing the incident field into different groups which are maximized at different times, relative enhancement can occur at different time. Indeed the number of local maxima can be increased to the point that the pulses overlap in time creating a roughly constant intensity over some stretch of time.

With regard to focusing at a point the correlation time is just the field-field correlation time, which is the correlation time of the incident pulse (Z Shi, OL 38, 271, 2013). But the correlation time could be longer for the total transmission. This question could readily be explored in simulations. There is no mention of the nature of possible mode mixing in the fiber. Is mode mixing absent or does it exist to a certain extent? It would be interesting to explore how changing the degree of mode mixing by bending or otherwise distorting the fiber would influence the result. It would seem that greater control over the pulse shape could be accomplished if there were mode mixing since all modes would be present at all times.

Since the energy emerging from the fiber has a considerable structure in space and time, it is not obvious what applications of a pulse with a large number of transverse modes. A discussion of this point would be important.

The grammar should be improved. For example:

-Our experimental setup enables to measure all... -- enables us to measure

-On the inset – In the inset

-are following the same trend -- following the same trend

-are being measured -- are measured

-are often measuring – experiments don't measure

-are adding –add

-on purpose – should strike this out

-A full control – full control

- More precisely, we have shown how to enhance or to attenuate the transmitted intensity at any arbitrary delay times, -- More precisely, we have shown how to enhance or attenuate the transmitted intensity at arbitrary delay times,

-authors should not be thanked for providing funds, only the funding agency should be mentioned

Reply to Reviewer #1

In this manuscript the authors demonstrated a full control of transmitted light intensities in all spatial and polarization modes of a multimode fiber. The multi-spectral transmission matrix was measured and a Fourier transform gave the time-resolved transmission matrix. The eigenstates of the time-resolved transmission matrix with the maximum/minimum eigenvalues were computed and launched into the fiber for temporal enhancement or attenuation of transmitted light at a specific delay time.

This method was already used in Ref. [19] for global focusing of light through a multimode fiber. The main improvement in this work is that both polarizations are measured at input and output of the fiber. Similar to Ref. [19], the temporal enhancement was shown indirectly by Fourier transform of the output field measured at each frequency.

Key point

This work and [19] are demonstrating different phenomena. [19] uses digital phase conjugation to recreate the spatial state of the source (focal spot) at a given delay. We find whatever spatial state maximises the response at a given time. [19] asks “what must we launch to create a focused spot at a given delay”, the performance of other delays is unspecified. We ask “what must we launch to have as much light as possible to arrive (or not arrive) at a given delay”

Elaboration

In Ref. [19], “Delivery of focused short pulses through a multimode fiber” by Morales-Delgado et al., the authors demonstrate digital optical phase conjugation (DOPC) of an ultrashort pulse through a multimode fiber (MMF), whereas we use transmission matrix methods. DOPC recreates states which physically existed previously. Transfer matrix methods allow us to find special spatial states in post-processing by analysis of the matrix, that have never been physically generated during the measurement. This opens up more possibilities for the kinds of states you can find.

In [19], the created output pulse is focused in space (one focal spot) and in time (see Fig R1 below) as the DOPC process recreates the image of the source on the other side of the MMF. Our work finds different types of phenomena. Rather than being spatially focused at a given delay, we find a more general class of propagation modes which arrive at a given delay, but have no constraints placed on them as to what they must look like spatially. Purely temporal focusing with spatial degrees of freedom being completely free to take whatever form gives the best temporal response.

[Redacted]

[19] has no attenuation nor polarization control. To our knowledge, no one has previously demonstrated the ability to perform attenuation at given delays, multiple pulses, and polarization control in the temporal domain over all spatial modes. The global enhancement in the delay domain was discussed in a different mode coupling regime in [35]. However:

- In [35] the MMF is in the strong mode coupling regime, while we are not (See Response to Question 1). We don't have any long range correlations in our results, which strongly differs from [35]. Their transmission matrix is isotropic, and they can use random matrix theory to predict the enhancement ratio, while we cannot apply it with our results.
- Their spectral bandwidth is very limited (6.4 nm while we have 110nm), which tends to average the temporal features of the output speckle in the delay domain.
- Polarization control, attenuation and multiple pulses are not demonstrated in [35].

It would be more impressive if the authors could directly measure the temporal pulse shape and show the temporal enhancement/attenuation, similar to what one of the authors had done for spatio-temporal focusing through disordered systems in Ref. [21].

Key point

This is a linear experiment. There would be no advantages to performing this experiment in the time-domain, and in fact a number of disadvantages. Frequency domain is generally the way to go if you've got access to a quality swept-source, and if you're not measuring non-linear phenomena. It's the reason so much characterisation equipment operates in the frequency domain. It gives you access to a lot of bandwidth and excellent dynamic range.

Elaboration

We measure over approximately 13.5 THz (110nm at 1565nm), equivalent to a sinc pulse of ~75 fs duration. This has allowed us to see very fine temporal features which to our knowledge have not previously been published. We also get full phase information (for both time and frequency domains) which allows us to find those special spatial states. In [21] we were looking at some non-linear phenomena (2 photon fluorescence) so we had to measure/demonstrate directly in the time-domain, but we also didn't have a reliable swept-source.

Below are my questions.

1. How strong were the mode coupling and the polarization mixing in the step-index fiber used in the experiment?

Key point

Short answer is "moderate" mode coupling. There's actually multiple regimes of mode coupling from weak to strong occurring simultaneously in the fibre between different sets of modes. We provide a processing of our transmission matrix to show that we are not in the strong mode coupling regime. The polarization mixing is not a meaningful concept in a circular symmetry waveguide particularly of this length, as all polarization bases are degenerate.

Elaboration

The question of "how strong is the mode coupling?" is a natural question to ask, however extracting the answer experimentally has a lot of nuance to it. In simulation the coupling strength if something you enter numerically, in experiment you're not privy to what's going on along the length of the fibre, only what you observe at the ends. Importantly, you also have to distinguish between genuine mode coupling inside the fibre, and "pretend" mode coupling, which is a result of your choice of measurement basis in the experiment. This topic has been discussed in [Carpenter et al., Laser and Photonics Review 2016]. In fact in a typical fiber, there's the dynamics of multiple strengths of mode coupling all occurring simultaneously between different groups of modes.

In the supplementary information, we show the amplitude of the transfer matrices (TM) either in the spectral or in the temporal domain in Fig. S1. We show in Fig. R2 below the TM at 1565nm. The matrix is in the linearly polarized mode basis, both at the input and at the output. The matrix is not diagonal, but not completely filled either.

Figure R2. Amplitude of the Transmission Matrix at 1565nm

There is no universal metric to describe mode coupling, particularly in the regime where modes are not completely coupled. We've chosen to use here the mode participation number MP per input mode based on the mode decomposition at the output, which quantifies how light is distributed over the different modes [Xiong et al., Optics Express 2017].

$$MP = \frac{(\sum_{i=1}^{N_{\text{output}}} |t_i|^2)^2}{N_{\text{output}} \times \sum_{i=1}^{N_{\text{output}}} |t_i|^4}$$

With N_{output} the number of modes at the output and (t_i) the coefficients of the mode decomposition of the i -th mode at the output. Fig. R3 shows MP for the different input modes. An isotropic TM would have a $MP > 0.5$. The average over all input modes is 0.16, which means we are in a mode coupling regime that is not "strong" [Xiong et al., Optics Express 2017].

Figure R3 Mode participation number at 1565 nm for every input mode

With regards to polarization, it's not particularly meaningful to think of "polarization mixing" in a waveguide with circular symmetry. There are only "modes", which have spatially dependent polarizations, not "polarization modes" and "spatial modes", they're really all mixtures of both. Again, there's a question of how to even define a polarization axis for a circular fiber. That's a tricky question unto itself, but one which was investigated previously in [Carpenter et al., Laser and Photonics Review 2016].

We've added the following sentence in the main document in Section 2, page 2 left column "A step index multimode fiber has more mode coupling and a more complex and spread impulse response than a graded index fiber with similar dimensions. **The multimode fiber we use in this experiment is not in the strong mode coupling regime [Xiong et al., Optics Express 2017] (see Supplementary material for more details on the mode coupling in the fiber)**". The above mode coupling discussion is also added as a new Supplementary material Section S2: "Mode coupling in the multimode fiber".

2. What determined the enhancement factor in Fig. 2b? Why it increased with the delay time? How did the enhancement factor depend on the fiber parameters and the input pulse width?

Key point

It is complicated to figure out what determines the enhancement factor, since the mode coupling is very complex and unpredictable. At long delays near cutoff it's possible to find modes which have experienced very little mode coupling and hence have naturally short temporal responses, which contribute to the increase of the enhancement factor. We provide additional processing of our experimental data to show the impulse response associated with the maximum eigenstate at every single delay, the corresponding field correlation to estimate the time width, and the spatial reconstruction of these modes.

Elaboration

What determines the enhancement factor is complicated by the fact that there's multiple different coupling regimes occurring simultaneously. In the strong coupling regime, where all modes are always coupled with all modes and the impulse response of all modes is basically the same: all modes arrive at all times. Hence to enhance at a given delay, you must interfere many of these broad temporal responses together to get small temporal features. So the enhancement relates to the number of modes N_{modes} that arrive at a given time. That's the regime common to scattering media like white paint.

On the other extreme, imagine a set of N uncoupled single-mode waveguides. The no-coupling regime. In this scenario, getting a sharp temporal response at the output requires you to couple to just a single waveguide. Each of the N modes arrives at the other end well focused in time as it ultimately only travelled 1 direct path from input to output.

In multimode fibers, both of these types of coupling regimes can be occurring simultaneously, and to different degrees amongst different modes, and at different delay times. At very long delays, which correspond to modes near cutoff, the propagation constants, group delays and spatial profiles start to change rapidly. This makes coupling more difficult. There are also no higher-order modes for them to couple to (only lower order modes), so any coupling to would-be higher-order modes just becomes loss rather than a broader temporal response. The core-cladding interface is also the most likely location for defects in a step-index fiber, which can also pull the propagation constants of would-be degenerate modes apart. We've seen similar phenomena previously in [Carpenter et al., Laser and Photonics Review 2016] and [Carpenter et al., Nat. Phot 2015], where the highest-order modes start to experience less mode coupling and get sharper temporal features, as they manage to travel from one end of the fiber to the other along a unusually direct path (like a ballistic photon). This does occur

at the expense of loss. You can see this in Figure R7 at the late delays, where the spatial patterns start to look like “textbook” modes, having traversed the fiber in nearly identical form like a traditional eigenmode.

We also stress that our experimental study shows results for a very common fibre (50 μm core, 0.22 NA). This study could be extended to other fibers (different refractive index distribution, different fiber parameters, different mode coupling regime etc...) or different scattering materials but that would be outside the scope of our manuscript. We show new control for the first time. Measuring the dependence of all properties as you sweep fibre parameters would be a paper itself. It gets very complicated, and you’d really need to be very accurately controlling the fibre manufacturing process. It’s also possible to measure two fibres, which ostensibly have the same fibre parameters, but can have quite different spatiotemporal dynamics. Some examples of that are in the supplementary of [Carpenter et al., Nat. Phot 2015]. It depends not only on the fibre parameters itself, but also on things like the acrylate coatings around the cladding, jacketing etc, as these influence microbending.

In the following, we provide additional processing of our data regarding the enhancement ratio as a function of delay.

In Figure R4, we plot the mode participation (as defined in the previous point) for each eigenstate maximizing the impulse response at a chosen delay. As expected, the number of modes at early and late delays is smaller than at mid delays, as demonstrated with principal modes [Xiong et al, Optics express 2017].

Figure R4 Mode participation max eigenstates

In Figure R5, we plot in a 2D graph the individual impulse response of the first max eigenstate at each delay time, as an extension to the plot of Figure 2a.

Figure R5 Impulse Response of the max eigenstate at 46 different arrival times

To estimate the time width of each individual impulse response, we calculate the square of the field correlation $|F_E(\Delta\tau)|^2$ in time [Z Shi et al, OL 2013], with :

$$F_E(\Delta\tau) = \frac{E(\tau_{max})E^*(\tau_{max} + \Delta\tau)}{(I(\tau_{max})I(\tau_{max} + \Delta\tau))^{1/2}}$$

With $E(\tau_{max})$ the reconstructed field constituted of the mode decomposition at delay τ_{max} , corresponding to the chosen enhanced delay. We plot $|F_E(\Delta\tau)|^2$ for all the impulse responses from Fig. R5 in Fig.R6.

Figure R6 Square of the field correlation function in time

At late delays, the temporal profiles of the achieved eigenstates are more sharply peaked compared to mid delays. This could be explained by the mode participation number and the lower mode coupling at these delays, as the modes are near cut-off [Carpenter et al., Laser and Photonics Review 2017; Carpenter et al., Nature photonics 2015].

Finally, we provide the spatial reconstruction at each delay for both polarization states in Fig. R7, as an extension to the inset of Fig. 2a. The modes at late delays are using the full aperture of the fiber and correspond to higher order modes that look like “textbook” eigenmodes, while the modes at early delays are mostly low order modes.

Spatial Reconstruction Pol. H

Spatial Reconstruction Pol. V

Figure R7 Spatial Reconstruction of the max eigenstate intensity images at different delays for both H (left hand side) and V (right hand side). The color-scale is identical for all the plots.

We've added in the main text in Section 3, page 4 left column "In Fig. 2b., we compare the enhancement

ratio $\eta(\tau)$ calculated with the experimentally measured impulse responses (red markers) with the expected enhancement $\eta^{\text{numerical}}(\tau)$ (green markers) for a set of different arrival times spread between $\tau = 0 \text{ ps}$ and $\tau = 45 \text{ ps}$. **The expected enhancement can be predicted with the TRTM. However we**

cannot use random matrix theory to predict the eigenvalue distribution as the MMF is not in the strong mode coupling regime [35]. The expected enhancement $\eta^{\text{numerical}}(\tau)$ is calculated with Eq. 2, where $I^{\text{numerical}}_{\tau_{\text{max}}}$ replaces $I_{\tau_{\text{max}}}$. As the TRTM contains the full spatiotemporal relationship between the input and the output fields, $\eta(\tau)$ follows, as expected, a similar trend as $\eta^{\text{numerical}}(\tau)$. Modes arriving at very long delays are higher-order modes, that are near cut-off. These higher-order modes experience less mode coupling and hence sharper temporal features [Carpenter et al., Laser and Photonics Review 2016], which tends to increase the enhancement ratio for the eigenmodes at long delays. Additional data processing, such as all the individual impulse responses used to calculate η and the spatial reconstruction of the beam at the different delays, are presented in the Supplementary Material."

We've added a new section **S4 in the Supplementary material "Extra data processing of the maximum eigenstate at different delay times"** that contains the above discussion.

3. In Fig. 3b, why the extinction ratio decreased with increasing delay time, opposite to the enhancement factor in Fig. 2b?

Key point

The extinction ratio is presented in log-scale, we're dealing with quite small differences. At long delays (higher-order modes) we're also working with effects like less experimental accuracy in the excitations and less power (worse signal-to-noise).

Elaboration

The numerical extinction ratio shows that in theory we are able to get a sharp zero intensity at every delay, as we can see in one example in Figure 3a and <20dB extinction ratio at all delays. However we cannot achieve experimentally this zero intensity as we've discussed in the manuscript in Section 4. The inaccuracy of the SLM mask and the experimental noise level are mostly responsible for it. The highest order modes are more spatially complicated and difficult to accurately excite. So even a small few percent imperfection in mode launch, which is still pretty good for arbitrary higher-order modes, could start to wash out the desired dips. This small amount of signal, as we can see on the red plot in the inset of Fig. 3a plays a more important role at large delays because the time of flight intensity gets smaller, which would tend to reduce the enhancement ratio at large delays. However we do not believe this "decreasing trend" is correlated with any trend in the enhancement ratio.

We've clarified this point in our manuscript In Section 4, page 4 right column:" *We plot the attenuation ratio in logarithm scale for a set of different delay time between 0 ps and 45 ps in Fig. 3b. The numerical extinction ratio <-20dB implies we should be able to completely attenuate the beam at every delay time, as the inset of Figure 3a illustrates for instance at specific delay τ_{min} . However, the experimental conditions are not allowing a pure zero-intensity in the experimentally measured impulse response (...) of a pure zero-intensity in the impulse response. Ultimately, this non-zero experimental intensity tends to decrease the extinction ratio at larger delays where the time of flight gets a smaller intensity."*

4. Figure 4 showed either horizontally or vertically polarized light was enhanced or suppressed at a specific delay time. Can the authors demonstrate arbitrarily polarization state, e.g., circular polarization, at a specific delay time, similar to what was done with continuous wave by Xiong et al. in Light: Science & Applications 7, 54 (2018)?

Key point

There's no significance to the H/V axis chosen, we could just as easily have optimised in any orthogonal polarization basis, and we've numerically demonstrated some examples below.

Elaboration

There's really no significance to linear vs. circular vs any other polarization basis for this experiment, they're just arbitrary experimental choices. If we'd put a quarter waveplate on the output of the MMF, we'd have been measuring and observing LCP/RCP on the camera instead. Below we've applied such a waveplate in post-processing to demonstrate what would occur. But ultimately, any polarization basis is about as good as any other for a circular fibre of this length, there's nothing particularly special about our choice of HV relative to the optical table it sits on. The polarization basis we chose is already arbitrary with respect to the fiber (it's just H and V relative to the camera/table, no relationship with the orientation of the fiber we're measuring).

In order to get a specific polarization state, we need to change the output polarization basis. Similarly to what was shown in the monochromatic regime in Xiong et al, LSA 2018, we can calculate the time resolved transmission matrix in a different output polarization basis, based on the experimentally measured matrix in the H/V basis. Eq S4 gives the polarization decomposition of a given time gated transmission matrix $U(\tau)$ at delay τ . In principle, the output polarization basis can be numerically rotated at will from $U(\tau)$. For instance, we want to access either the left hand or the right hand output polarization states. From these components we can write the matrix $U_{HR}(\tau)$ which is the matrix relating the H input on the SLM to the right hand polarization state at the output: $U_{HR}(\tau) = 1/\sqrt{2} (U_{HH}(\tau) - iU_{HV}(\tau))$. We can calculate also the full matrix on the left hand and right hand circular output basis: $[U_{HR}(\tau) \ U_{VR}(\tau); U_{HL}(\tau) \ U_{VL}(\tau)] = 1/\sqrt{2} [1 \ -i; 1 \ i] [U_{HH}(\tau) \ U_{VH}(\tau); U_{HV}(\tau) \ U_{VV}(\tau)]$ with $U_{ij}(\tau)$ the sub time gated matrix at delay τ connecting the i -th input polarization state to the j -th output polarization state. This process is then equivalent to adding a quarter wave plate at 45 degrees of the horizontal axis onto the output. If we want to enhance the right hand circular polarization state, then we can use the maximum eigenstate of $U_R(\tau) = [U_{HR}(\tau); U_{VR}(\tau)]$. A similar approach can be set with the control of the left hand output state, by replacing $U_R(\tau)$ with $U_L(\tau) = [U_{HL}(\tau); U_{VL}(\tau)]$.

Furthermore, we can also access the linear +45 / -45 degrees from the time resolved transmission matrix in the H/V basis using: $[U_{H-45}(\tau) \ U_{V-45}(\tau); U_{H+45}(\tau) \ U_{V+45}(\tau)] = 1/\sqrt{2} [1 \ -1; 1 \ 1] [U_{HH}(\tau) \ U_{VH}(\tau); U_{HV}(\tau) \ U_{VV}(\tau)]$.

Note that we could apply a similar method (e.g. change of the output polarization basis) on the multi spectral matrix directly. Instead of measuring H/V on the camera, we could be measuring the left hand circular and the right hand circular polarizations states on both sides of the camera with a quarter wave plate in the experiment, or by applying a numerical quarter wave plate on the output. Or +45 degrees / -45 degrees by applying a rotation matrix on the output. Enhancing only a given polarization state (e.g. either LCP or RCP, or +45 or -45) would then be equivalent to enhancing one side of the camera, as we did for either H or V in Fig. 4 of the manuscript.

We demonstrate in Fig. R8 how to enhance the total transmission with either left hand polarization or right hand polarization, or linear polarization at either +45 degrees or -45 degrees, at delay $\tau=15$ ps. We use here a numerical propagation, as described in the main manuscript in section 3, to obtain the polarization and time resolved output fields. We then apply a phase shift ϕ (16 steps between 0 and 2π) for both H and V to look at the output polarization state for all the different spatial positions. The total intensity is shown on a grey scale colour-map that is identical for all the subfigures. The polarization state is represented as a vector, with the size of the arrow related to the amplitude of H and V, and its direction related to the phase difference between H and V. The H axis is set as horizontal, and the V axis as vertical. The top line corresponds to enhancing either H or V at the output, which is demonstrated in Fig. 4 of the main manuscript. The middle line shows the enhanced field with either a left hand or a right hand polarization state, and the bottom line shows both 45 degrees linear polarization states. The expected helicity is observed, as well the expected angle for the linear polarization. Note that a only a phase shift has been applied when calculating $U_L(\tau)$ and $U_R(\tau)$, which

explains why we don't have a perfect circular state, as the power ratio between H and V differs in all spatial positions.

Figure R8 (Numerical propagation) Intensity images upon enhancing different output polarization states at $\tau=15\text{ps}$ with the time resolved transmission matrix. Both the H intensity and the V intensity are superimposed. Same grey level intensity map for all plots. The arrows represent the polarization vector upon a phase shift of 0 (red arrow) to 2π (green arrow) on both H and V components Top left: H; Top right: V; Mid Left: Left hand; Mid right: Right hand. Bottom left: Linear 45 degrees; Bottom right: Linear -45 degrees

The following discussion has been added in the Supplementary information as a new Section S6: “Arbitrary polarization state at a chosen delay”. We also add the following sentence in Section 5 of the main manuscript: “Fig. 4c and Fig. 4d unambiguously reveal a strong attenuation at τ_s for the

controlled output polarization state. In principle, any arbitrary polarization state can be achieved at a given delay, similarly to previous work in the monochromatic regime [Xiong et al., Light Sci appl 2018], based on the superposition of the four sub time gated transmission matrices of Eq.4. Some examples are shown in Supplementary Information. ”

5. When the transmitted light intensity is enhanced at multiple delay times in Fig. 5, how much was the total enhancement factor? Was it lower than the enhancement factor at a single delay time?

Key point

We're not exactly sure what you mean by "total enhancement factor", as our enhancement ratio is defined only at a single delay per conservation of energy.

Elaboration

The enhancement we use in Eq. 2 is defined only at a single delay and not over the total temporal duration/bandwidth, that is different from the speckle/focus contrast definition in scattering media [Shi et al, OL 2013; Mounaix et al., Optica 2017]. In our experiment, the output power is almost the same whatever is being displayed on the SLM as per conservation of energy; therefore there is not really a total enhancement along the temporal duration. However, we can compare the enhancement at a single delay to the enhancement at multiple delays.

- The enhancement at multiple delays is calculated with the impulse response from Figure 5. We choose for instance to perform the analysis with Fig. 5a, which contains $N_{\text{superpos}}=6$ peaks within 40ps. N_{superpos} enhancement ratios can be extracted, as defined in Eq. 2 of the main manuscript, at the N_{superpos} peaks. These N_{superpos} enhancement ratios are plotted in brown in Figure R9.
- As a comparison, we extract the enhancement ratios at single delays (with the same arrival time as in Fig. 5a) from Fig. 2b. The corresponding enhancements are plotted in different colours from red to orange in Figure R9.

As expected, the enhancement at multiple delays (e.g the brown crosses in Fig R9) is lower per delay, when we compare to a single delay enhancement. We can see that if we multiply the "brown" points of Fig. R9 by N_{superpos} we get a "higher" enhancement. Indeed, when looking at the impulse responses of Fig. 5a and Fig. 2a, upon multiple delay control there is less "background" outside the uncontrolled delays compared to a single delay control. The output energy is more controlled upon multiple delay as we are using more modes, but the price to pay is more loss due to the more complex mask

superposition on the SLM, which induces extra loss. However, we cannot define a total enhancement due to conservation of energy.

Figure R9 Enhancement ratio when enhancing either a multiple delay solution (brown colour, from the impulse response shown in Fig. 5a) or a single delay (red to orange colour, one point corresponds to a single delay enhancement extracted from Fig. 2b)

To clarify our manuscript, we added the following sentence in Section 6, page 6: “We clearly identify $N_{superpos}$ peaks in Fig. 5a and Fig. 5b. **The enhancement ratios at the $N_{superpos}$ delays are much lower than the individual delay enhancement ratios presented in Fig. 2b. However the total transmitted power remains similar for either a single delay enhancement or a multiple delay control. The amplitude ratio between these peaks could ...**”

In summary, the experimental studies presented in this paper were solid, but physical understanding of the experimental results was missing.

We hope we have provided more physical explanations, data processing and additional results to clarify our manuscript.

Reply to Reviewer #2

Control of the temporal and polarization response of a multimode fiber

This article describes the control of the temporal shape and polarization of the total transmission through a multimode fiber. This is in contrast to typical pulsed experiments in which the pulsed intensity is controlled in a single speckle spot in transmission through a scattering medium. In that case, the total transmitted power is dominated by the background, which is enhanced to a lesser degree than the intensity in the focal spot. Here the total transmission over the full core of the fiber is shaped. In steady state as well as in pulsed experiments, the ratio between transmission in the highest and lowest transmission eigenvalues is determined by the completeness of the transmission matrix [S M Popoff, PRL, 112, 133903, 2014; A Goetschy PRL 111, 063901 2013]. Apparently, the matrix

is quite complete here giving a large ratio of maximum to minimum intensity at a specified time. To accomplish this it is important to have full control over the polarization. A detailed discussion of the extent and sources of incompleteness of the transmission matrix in this case would be of interest.

Key point

One of the nice aspects of working with a multimode fibre is that the transmission matrix can be completely characterised. Compared to white paint or biological tissue, MMF has a relatively small number of well-defined modes, and we can measure all of them.

Elaboration

In contrast with a multiple scattering material such as white paint (as in S M Popoff, PRL, 112, 133903, 2014 and A Goetschy PRL 111, 063901 2013), we can experimentally launch and measure all the $N_{\text{input}}=254$ propagating modes of our MMF. Each input mask is calculated with a Gerchberg Saxton algorithm as described in the supplementary material, which enables us to launch every propagating mode in the fiber. To ensure we measure all the information at the output, our digital holography mode decomposition has a higher number of output modes $N_{\text{output}}=650$ modes. Therefore, our transmission matrix is “complete”.

We clarified this point in our manuscript in Section 2: *“U is a $N_{\text{input}} \times N_{\text{output}}$ matrix, with N_{input} and N_{output} the number of spatial and polarization modes at the input and at the output of the system. **Note that in a multimode fiber, all the N_{input} modes (on the order of 100-1000 modes depending on its geometrical parameters) could be measured in contrast with highly disordered materials [S M Popoff, PRL, 112, 133903, 2014; A Goetschy PRL 111, 063901 2013]. The transmission matrix has been widely ...”** and *“We use a swept-wavelength interferometer (SWI) to measure the multi spectral transmission matrix of the MMF [27, 30]. **We measure the complete transmission matrix by generating all the spatial and polarization modes (a total of $N_{\text{input}} = 254$ modes) on the input facet of the MMF using the SLM.**”**

Unlike the transmission through a scattering medium, it appears that all channels are completely transmitted and so the number of channels is just the number supported by the fiber. By dividing the incident field into different groups which are maximized at different times, relative enhancement can occur at different time. Indeed the number of local maxima can be increased to the point that the pulses overlap in time creating a roughly constant intensity over some stretch of time. With regard to focusing at a point the correlation time is just the field-field correlation time, which is the correlation time of the incident pulse (Z Shi, OL 38, 271, 2013). But the correlation time could be longer for the total transmission. This question could readily be explored in simulations.

Key point

There is no correlation time in our experiments as we use a frequency domain characterization technique. We provide additional simulation results of the propagation of a short pulse and we calculate the numerically propagated field correlation in the temporal domain, as well as the enhancement ratio.

Elaboration

We do not have a correlation time of the incident pulse in our experiment. We use swept wavelength digital holography which is a frequency-domain technique. It corresponds to a sinc pulse of ~ 75 fs, in contrast with [Shi et al, OL. 2013] where a Gaussian input pulse is sent through a 1D disordered material. We use the field correlation defined in [Shi et al, OL. 2013] to extract the time width of the impulse response, as shown in Fig. R6.

We provide here some simulation results, based on the experimentally measured MTM, of the impulse response we would measure with a short input pulse. We then calculate the enhancement ratio, as well as the field correlation in the temporal domain.

A Gaussian short pulse in the time domain of given duration corresponds to a Gaussian amplitude distribution in the spectrum with a flat spectral phase. We thus filter the spectral amplitude of our experimentally measured MTM with such Gaussian amplitude along the wavelength axis. We numerically propagate the input field with this spectrally filtered MTM and get the spectrally resolved output field. A Fourier transform along the frequency axis gives the time resolved output field upon numerical propagation of the input pulse, from which we get the impulse response. We can then extract the enhancement factor as a function of delay time for different input pulse durations, as well as the field correlation.

Figure R10 Simulation of numerical propagation of an ultrashort pulse of given duration. We show results for two different input pulse durations: (a,b,c) 260fs and (d,e,f) 1.3ps. (a,d) Impulse response when enhancing at $\tau_{\max}=14\text{ps}$ (b,d) square of the field correlation in the temporal domain (c,f) enhancement ratio

In Fig. R10 we plot the impulse response of the maximum eigenstate at a single delay, for instance $\tau_{\max}=14\text{ps}$, for either an input pulse of duration 260fs (Spectral Bandwidth $\approx 20\%$ of our full spectral measurement) in Fig. R10a or 1.3ps (Spectral Bandwidth $\approx 3\%$ of our full spectral measurement) in Fig. R10d. The temporal features of the background outside τ_{\max} , as well as the width of the peak at τ_{\max} have been averaged out compared to Fig. 2a, as they have now an average width that is the input pulse duration. To confirm this, we plot the field correlation in the time domain in Fig R10b and Fig R10e for both input pulse durations. In contrast with Fig. R6, the time width of the peak of the impulse response seems constant for the different delays and identical to the input pulse duration, as studied in [Z Shi et al., OL 2013] for spatio-temporal focusing in a single position of an input Gaussian pulse. We finally plot the enhancement ratio with the same arrival times as in Fig. 2b in Fig. R10c and R10f. There is no clear trend for enhancement ratio as a function of the input pulse width for our MMF in this mode coupling regime. As mentioned above, there are dynamics of both strong coupling and weak coupling occurring amongst different groups of modes/delays in the fibre, and it can be quite sensitive to very specific details of the fibre parameters.

The comparison of the global focusing and spatio-temporal focusing (as in [Shi et al. OL 2013, Mounaix et al., PRL 2016]) could be further studied in different mode coupling regimes in a later study.

We've added the above discussion as a **new section S4.E "Simulation with an input pulse of given duration" in the Supplementary material.**

We also added in the main document in Section 3: "As the TRTM contains the full spatiotemporal relationship between the input and the output fields, $\eta(\tau)$ follows, as expected, a similar trend as $\eta^{numerical}(\tau)$. **Modes arriving at very long delays are higher-order modes, that are near cut-off. These higher-order modes experience less mode coupling and hence sharper temporal features [Carpenter et al., Laser and Photonics Review 2016], which tends to increase the enhancement ratio for the eigenmodes at long delays. Additional data processing, such all the individual impulse responses used to calculate η and the spatial reconstruction of the beam at the different delays, are presented in the Supplementary Material. Finally, in our study we do not send an input pulse as we measure the data with swept wavelength digital holography. A simulation of the propagation of an input pulse, and its effect on the impulse response and the enhancement ratio is presented in the Supplementary material."**

There is no mention of the nature of possible mode mixing in the fiber. Is mode mixing absent or does it exist to a certain extent? It would be interesting to explore how changing the degree of mode mixing by bending or otherwise distorting the fiber would influence the result. It would seem that greater control over the pulse shape could be accomplished if there were mode mixing since all modes would be present at all times.

Key point

There are both strong and weak coupling occurring within different groups of modes/delays in the fibre.

Elaboration

As shown in Fig. R1, there is clearly mode coupling as the matrix is not diagonal in the LP mode basis, and the mode participation number is much greater than 1 as shown in Fig. R2. However, the matrix is not isotropic as the mode participation number is <0.5 , thus we are not in the strong mode coupling regime. All the results in the manuscript have been measured in this regime, from a standard MMF. The degree of control in another regime (e.g. stronger mode coupling or in a highly scattering material) and the comparison with our regime would be out of the context of our manuscript (which shows such control for the first time) and could be part of a later study.

Since the energy emerging from the fiber has a considerable structure in space and time, it is not obvious what applications of a pulse with a large number of transverse modes. A discussion of this point would be important.

Key point

In an analogous fashion to phenomena that are spatially focused, but not temporally focused, these states can be used for applications requiring high temporal resolution through scattering. Or where spatial coherence must be maintained over a broad bandwidth.

Elaboration

The ability to control the global temporal response over a large number of transverse modes could find applications in temporal focusing microscopy [D. Oron et al, Opt. Express (2005), Zhu et al., Optics Express 2005]. Indeed, the temporal degrees of freedom are exploited rather than spatial degrees of freedom, and have been used for in depth multiphoton imaging [Adrià Escobet-Montalbán et al, Science Advances Vol. 4, no. 10, eaau1338 2018]. Our method could also be used for polarization

sensitive optical coherence tomography (PS-OCT) [Li et al, Optics express 2015; Walther et al. Biomedical optics Express 2019 ;de Boer et al., Biomedical Optics Express 2017,].

We implemented these additional applications in the conclusion section 7: *“This work opens interesting perspectives in applications that require the measurement or control of spatio-temporal properties of light in scattering media. A pulse with a large number of transverse modes could be useful for the following non exhaustive list of examples: the study of correlations in disordered systems [32, 43–45] telecommunications [46], non-linear imaging [29, 47] and endoscopy [48–50] through complex systems, time-gating imaging [51, 52], temporal focusing microscopy [D. Oron et al, Opt. Express 2005, Zhu et al., Optics Express 2005, Adrià Escobet-Montalbán et al, Science Advances 2018], and high power fiber lasers [53]. Our polarization control of the beam in the temporal domain could find applications in polarization sensitive optical coherence tomography [Li et al, Optics express 2015; Walther et al. Biomedical optics Express 2019 ;de Boer et al., Biomedical Optics Express 2017].”*

The grammar should be improved. For example:

-Our experimental setup enables to measure all... -- enables us to measure

-On the inset – In the inset

-are following the same trend -- following the same trend

-are being measured -- are measured

-are often measuring – experiments don’t measure

-are adding –add

-on purpose – should strike this out

-A full control – full control

- More precisely, we have shown how to enhance or to attenuate the transmitted intensity at any arbitrary delay times, -- More precisely, we have shown how to enhance or attenuate the transmitted intensity at arbitrary delay times,

-authors should not be thanked for providing funds, only the funding agency should be mentioned

We thank the Referee for the careful reading of our manuscript. We have implemented the grammatical corrections in the revised manuscript.

Reviewers' Comments:

Reviewer #1:

Remarks to the Author:

The authors have addressed all my questions and modified their manuscript accordingly. I recommend the publication of the current version in Nature Communications.

Reviewer #2:

Remarks to the Author:

I believe the revised manuscript satisfactorily addresses the questions raised by the referees with apt reasoning and analysis. Key issues related to the control of propagation through multimode optical fibers of importance in telecommunications and sensing are analyzed in an informative fashion. I therefore recommend the manuscript be published in Nature Communications.

REVIEWERS' COMMENTS:

Reviewer #1 (Remarks to the Author):

The authors have addressed all my questions and modified their manuscript accordingly. I recommend the publication of the current version in Nature Communications.

We thank the Referee.

Reviewer #2 (Remarks to the Author):

I believe the revised manuscript satisfactorily addresses the questions raised by the referees with apt reasoning and analysis. Key issues related to the control of propagation through multimode optical fibers of importance in telecommunications and sensing are analyzed in an informative fashion. I therefore recommend the manuscript be published in Nature Communications.

We thank the Referee.